# Anomalous anti-Kasha excited-state luminescence from symmetry-breaking heterogeneous carbon bisnanohoops

Xinyu Zhang[1,4], Cheng Chen[2,4], Wen Zhang[1], Nan Yin[1], Bing Yuan[1], Guilin Zhuang [3], Xiao-Ye Wang [2] & Pingwu Du [1] ✉

It is a long-standing scientific controversy to achieve anti-Kasha-type multiple emissions by tuning the structures at a molecular level. Although it is known that some conjugated structures have excitation-dependent multiple emissions, no all-benzenoid molecules have yet been reported, the emissions of which originate from different excited states. Herein, we report the design of two symmetry-breaking heterogeneous carbon bisnanohoops that in solution become multiple fluorescent emitters with unusual anti-Kasha characteristics. This phenomenon can be spectroscopically and theoretically explained and will find applications in a wide range of sensing and imaging technologies.

Kasha's rule, a general principle in basic spectroscopy, states that the emitting level of a given multiplicity is the lowest excited level of that multiplicity, which was formulated by Michael Kasha in 1950[1]. Most fluorescent molecules follow this rule very well, but very few molecules are reported to break this rule, such as azulene and thioketones (Figs. 1a, b)[2-7]. Azulene only shows anti-Kasha $S_2$ fluorescence due to a large energy separation between the $S_2$ and $S_1$ states ($\Delta E(S_2 - S_1) \sim 14,000 \text{ cm}^{-1}$), leading to a relatively slow internal conversion of $S_2 \rightarrow S_1$, and $S_2$ radiative emission rates can compete favorably with internal conversion (Fig. 1d)[3,4]. Some thioketones, such as thiophosgene[5] and xanthione (Fig. 1b)[6], have a larger oscillator strength of $S_2$ than that of $S_1$ and demonstrate fluorescent emission from the $S_2$ excited state. In those molecules, the $S_1 \rightarrow S_0$ transition is negligible (Fig. 1d)[8]. The anti-Kasha emission provides possibilities to control the excited-state transformations for a variety of fundamental research and practical applications[9-14]. In the past few decades, a few other molecular systems, such as phenanthroimidazole-type dyes[15], tricarbocyanine[16], cycl[3.3.3] azines[17], and triphenylmethane[18,19], have also been reported to show anti-Kasha emissions, but many examples are still under debate[20]. Moreover, almost all the reported organic molecules that violate Kasha's rule are non-benzenoid aromatic hydrocarbons or

heteroatom-containing organic molecules, regardless of whether they are a definite anti-Kasha molecular system or a controversial molecular system[20-23]. To date, no such molecule containing only benzenoid moieties has been reported to have multiple fluorescent emissions with unusual anti-Kasha characteristics.

Benzenoid-type luminogens have the advantages of wide variety, dynamic tunability, rich physical properties, and excellent stability, which are attractive for many applications[24-29]. The conjugated backbones of benzenoid macrocycles support delocalized electronic excitations, resulting in interesting tunable dynamics by changing cyclic geometries and/or symmetries[30,31]. Conjugated cycloparaphenylene (CPP) nanohoops, representing the shortest sidewall segments of carbon nanotubes, have demonstrated intriguing luminescent and electronic properties[31-35]. Due to a conservation of orbital symmetry for the centrosymmetric CPPs, the HOMO-LUMO optical transitions are Laporte forbidden[36]. As a result, the photophysical properties of CPP differ significantly from those of linear conjugated materials. CPPs have virtually identical absorbance maxima and unique fluorescence emissions from spatially localized $S_{1'}$ states to the ground state $S_0$[37]. In 2019, Jasti and coworkers discovered that breaking the centrosymmetric CPP can alter fluorescent properties, although no multiple emissions were observed[38].

[1]Hefei National Research Center for Physical Sciences at the Microscale, Key Laboratory of Precision and Intelligent Chemistry, Department of Materials Science and Engineering, University of Science and Technology of China, 96 Jinzhai Road, Hefei, Anhui Province 230026, China. [2]State Key Laboratory of Elemento-Organic Chemistry, College of Chemistry, Nankai University, Tianjin 300071, China. [3]College of Chemical Engineering, Zhejiang University of Technology, 18 Chaowang Road, Hangzhou 310032, China. [4]These authors contributed equally: Xinyu Zhang, Cheng Chen. ✉e-mail: dupingwu@ustc.edu.cn

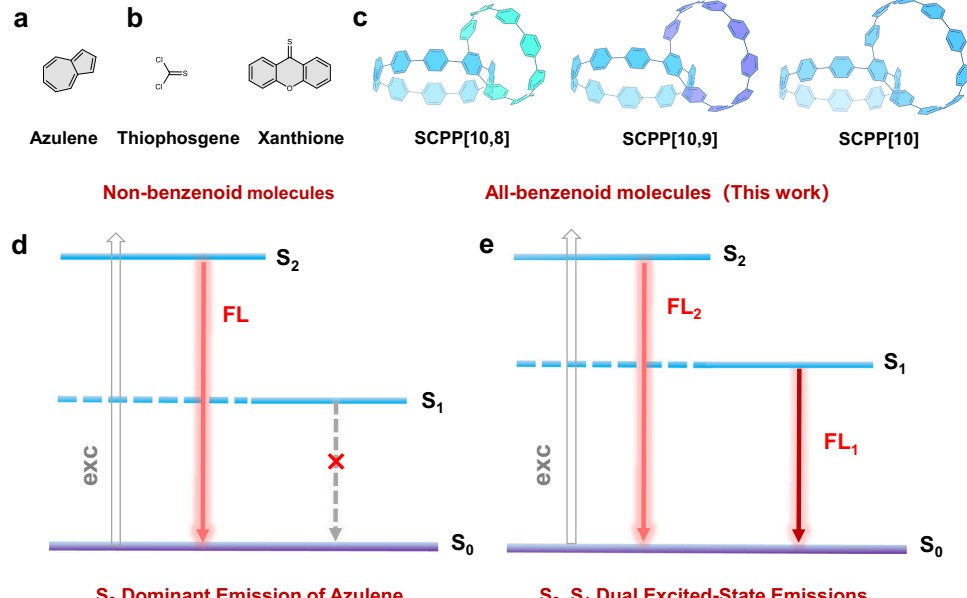

**Fig. 1 | Schematic representation. a** Structure of azulene. **b** Structures of thio-phosgene (left) and xanthione (right). **c** Structures of heterogeneous bisnanohoops **SCPP[10,8]** (left), **SCPP[10,9]** (middle) and homogeneous bisnanohoop **SCPP[10]** (right). **d** Anti-Kasha emission of azulene. **e** $S_2$, $S_1$ dual excited-state emissions of **SCPP**s bisnanohoops.

Herein, we report a distinctive type of symmetry-breaking heterogeneous carbon bisnanohoops, (1,4)[10,9]cycloparaphenylenophane (**SCPP[10,9]**) and (1,4)[10,8]cycloparaphenylenophane (**SCPP[10,8]**) (Fig. 1c), achieving excitation-dependent multiple emissions from the upper and lowest singlet excited states with unusual anti-Kasha characteristics. **SCPP[10,9]** and **SCPP[10,8]** have two different CPP-based nanohoops that are fused in a central benzene. Bisnanohoops with heterogeneous CPP-based macrocycle compositions are expected to have extraordinary photophysical phenomena derived from breaking the symmetry of the original centrosymmetric structure. The homogeneous bisnanohoop **SCPP[10]** was also studied for comparison (Fig. 1c)[26]. By adjusting the size of the bisnanohoops and tuning the balance between the high-energy excited state and low-energy excited state, the interesting excitation-dependent multicolor luminescence and systematic photophysical properties of bisnanohoops were investigated. This work demonstrates an intriguing example of organic all-benzenoid molecules with $S_2$ and $S_1$ dual excited-state emissions and wide-range emissions spanning over 150 nm (Fig. 1e).

## Results

### Synthesis and characterizations of SCPPs

The key to the synthesis of heterogeneous bisnanohoops was the use of ridged 1,4-syn-dimethoxy-2,5-cyclohexadienes (**1** and **2**) as masked benzene rings to react with ditriflate [10]CPP single nanohoop **3**. This appropriately substituted cyclohexadiene unit is able to alleviate strain and allow for Suzuki-Miyaura cross-coupling/macrocyclization to bisnanohoops with varying sizes. In the final reductive aromatization step, these cyclohexadiene-containing macrocycles were treated with $H_2SnCl_4$ at room temperature to achieve the fully benzenoid bisnanohoops **SCPP[10,9]** and **SCPP[10,8]** (Fig. 2). The successful synthesis of heterogeneous bisnanohoops can be supported by different characterization techniques, including various NMR spectra and high-resolution mass spectrometry (HR-MS) (Supplementary Figs. 1–20).

### Anti-Kasha emissions of SCPPs in solution

The photophysical properties of **SCPP[10,8]**, **SCPP[10,9]**, and **SCPP[10]** were investigated in dilute dichloromethane (DCM) solutions. As shown in Fig. 3a–c, the UV-Vis absorption spectra of **SCPP**s

exhibited broad absorption bands in the region of 280–480 nm. The maximum absorption peaks appear at 359 nm, 356 nm, and 351 nm for **SCPP[10,8]**-**SCPP[10]**, respectively. In addition, there are an obvious shoulder peak at approximately 340 nm and a weak shoulder peak in the longer wavelength region (at ~400 nm) for all **SCPP**s.

Furthermore, the steady-state photoluminescence (PL) spectra of **SCPP[10]**, **SCPP[10,9]**, and **SCPP[10,8]** were preliminarily measured under excitation at 360 nm (Fig. 3a–c). Each **SCPP** has a size-dependent low-energy emission (520 nm for **SCPP[10]**, 528 nm for **SCPP[10,9]**, and 545 nm for **SCPP[10,8]**) and two high-energy shoulder peaks at around 470 nm and 390 nm.

Unexpectedly, distinctive excitation-dependent multicolor fluorescence under different excitation wavelengths was clearly observed in their photoluminescent spectra and photographs (Fig. 3d–f). Under irradiation at 300 nm UV light, **SCPP[10]** and **SCPP[10,9]** displayed green fluorescence, and **SCPP[10,8]** showed blue fluorescence. Under irradiation at 365 nm UV light, **SCPP[10]** emitted light yellow fluorescence, **SCPP[10,9]** emitted yellow fluorescence, and **SCPP[10,8]** was remarkably orange-emissive. The emission color of **SCPP**s can be reversibly tuned through UV 365/300 nm switch processes, showing excellent sensitivity and reversibility. These processes were also monitored by photoluminescent spectra.

To further understand these anomalous phenomena, the detailed excitation wavelength-dependent fluorescence spectra of **SCPP**s were further measured (Fig. 3g–i). When the excitation wavelength ranged from 300 to 400 nm, these **SCPP**s demonstrated interesting multiple emissions. For instance, when a short excitation wavelength (<340 nm) was applied to **SCPP[10,8]**, two emission bands maximized at 392 nm and 467 nm were observed, while the latter dominated the fluorescence spectra. However, when the excitation wavelengths were slowly changed from 340 to 400 nm, a lower-energy emission band maximized at ~545 nm clearly appeared and gradually became the major emission, accompanied by a decrease in the higher-energy emission bands. The wide-range wavelength shift (155 nm) in the emission spectrum is consistent with the remarkable fluorescent color change of **SCPP[10,8]** from blue to orange. Interestingly, **SCPP**s exhibited similar high-energy fluorescence emission band maximized at ~390 nm and significantly

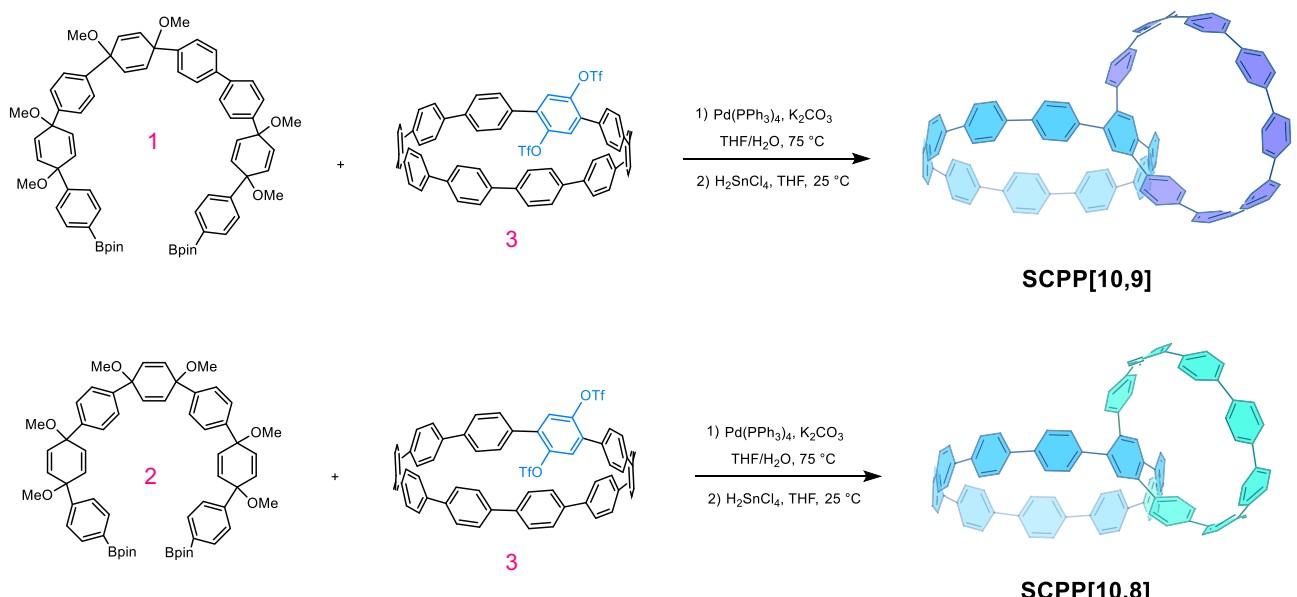

**Fig. 2 | Synthesis procedure for heterogeneous bisnanohoops.** Synthesis strategies of heterogeneous bisnanohoops **SCPP[10,9]** and **SCPP[10,8]**.

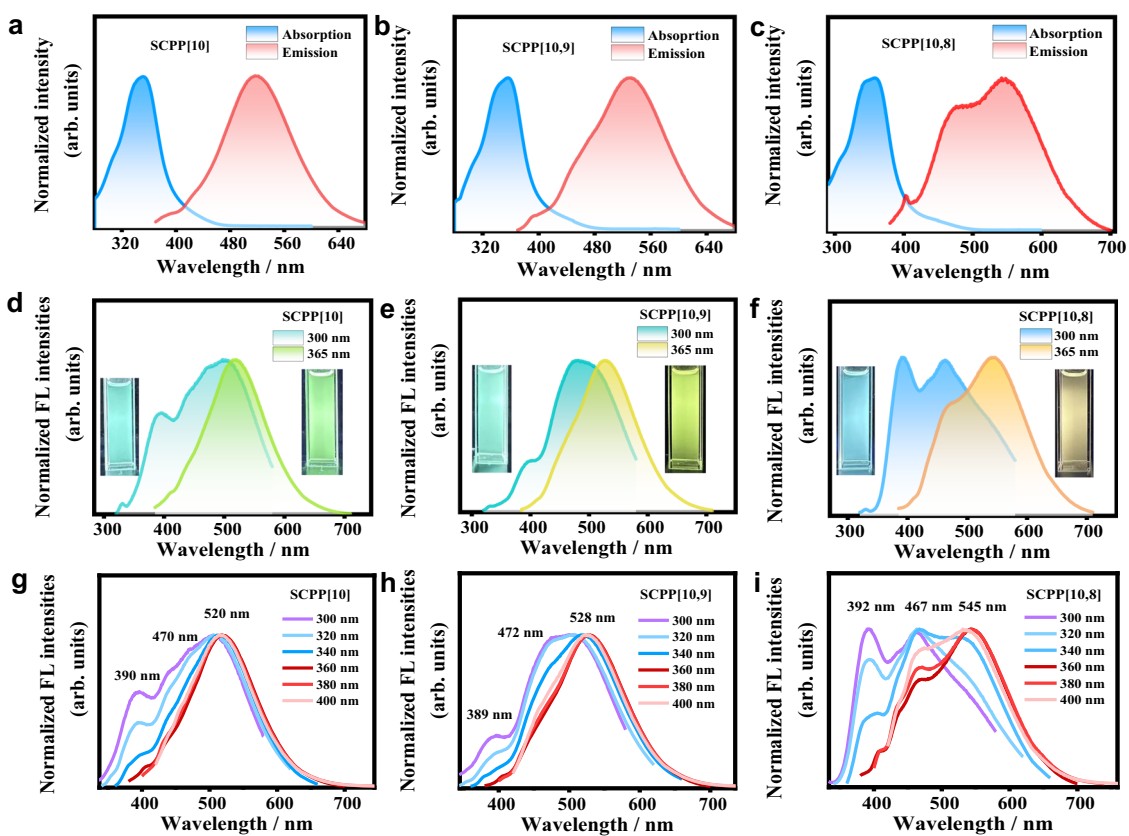

**Fig. 3 | Anti-Kasha emissions of SCPPs in solution.** Absorption and emission spectra (under excitation at 360 nm) of **SCPP**s: (**a**) **SCPP[10]**; (**b**) **SCPP[10,9]**; and (**c**) **SCPP[10,8]**. Fluorescence photographs and emission spectra of **SCPP**s under 300/365 nm UV light in dilute DCM ($c = 5 \times 10^{-6}$ mol/L): (**d**) **SCPP[10]**; (**e**) **SCPP[10,9]**; and (**f**) **SCPP[10,8]**. Excitation-dependent emissive properties of **SCPP**s: (**g**) **SCPP[10]**; (**h**) **SCPP[10,9]**; and (**i**) **SCPP[10,8]**.

different low-energy emission bands, in which the maximum peaks are located at 520 nm for **SCPP[10]**, 528 nm for **SCPP[10,9]**, and 545 nm for **SCPP[10,8]**. The relative proportions of different emission bands are directly related to the excitation wavelength, with higher energy excitations leading to increased emissions at shorter wavelengths. These results strongly indicated that the emissions come from more than one excited state. Besides, fluorescence quantum yields for the different emission processes were measured in a dilute solution at room temperature (Supplementary Table 1). The quantum yields from the low energy emission of **SCPP**s decreased as bisnanohoops got smaller, which is similar to the size-dependent emission properties of CPPs.

## Solid-state emissions of SCPPs

The solid-state fluorescence properties of **SCPP**s have also been studied. As observed with the naked eye, the color of **SCPP**s gradually changed from light yellow to orange as the ring size decreased (Supplementary Fig. 21). All these solid samples exhibited a single and bright fluorescence under a handheld UV light, and their emissions did not change with different excitation wavelengths (Fig. 4a, c and Supplementary Fig. 22). To further verify this phenomenon, the PL spectra of **SCPP** films were measured, and the results confirmed the independence of excitation (Fig. 4b, d and Supplementary Fig. 23). Therefore, **SCPP**s exhibit excitation-dependent multiple emission properties in solution, which violates the Kasha's rule. However, in the solid state, the high energy emissions vanished and only showed the lowest energy emissions, which obeys the Kasha's rule (Fig. 4e).

## Theoretical calculations of SCPPs

The subtle interactions of geometry, symmetry, and strain plays a significant role in many interesting properties of these **SCPP**s. Density functional theory (DFT) calculations were performed at the theoretical level of B3LYP/6-31 G(d, p) utilizing the Gaussian 16 program to investigate these basic properties of **SCPP**s[39]. Geometrical structure optimization showed that **SCPP**s feature $C_2$ symmetry, and the symmetry of bisnanohoops is obviously reduced compared with that of $D_2$-symmetric single nanohoop [10]CPP[31]. The twist angle of the bridging phenyl is 12.74° for **SCPP[10,8]**, 11.53° for **SCPP[10,9]**, and 10.05° for **SCPP[10]**. As the ring size decreases, the torsion of bridging phenyl gradually increases. Moreover, the strain energies were estimated to be 110.59 kcal/mol for **SCPP[10]**, 118.95 kcal/mol for **SCPP[10,9]**, and 130.98 kcal/mol for **SCPP[10,8]** (Supplementary Fig. 24 and Supplementary Table 2). Strain is a critical factor that endows nanohoops with unusual photophysical properties[31]. As the diameter decreases, these

**SCPP**s have increasing strain energies, changing the geometry and optical properties.

## Experimental and computational verification of anti-Kasha emission

To further explore the anti-Kasha properties of **SCPP**s, the absorption spectra were further investigated by theoretical calculation and compared with excitation spectra (Fig. 5a–d). Frontier molecular orbital analyses of **SCPP**s were performed to provide insight into the slight changes in absorption peaks. Time-dependent density functional theory (TDDFT) calculations revealed that the high-energy absorption bands of **SCPP**s are assignable to the $S_0 \rightarrow S_4$ (HOMO → LUMO + 3 and HOMO - 3 → LUMO) transition, and the relatively low-energy absorption bands of **SCPP**s are assignable to the $S_0 \rightarrow S_3$ (HOMO - 1 → LUMO + 1, HOMO - 2 → LUMO, and HOMO → LUMO + 2) transition, and the weak shoulder peak in the longer wavelength region (at ~400 nm) are assignable to $S_0 \rightarrow S_1$ (HOMO → LUMO) (Fig. 5d, Supplementary Figs. 25–28 and Supplementary Tables 3–6). As the ring size increases, the energy level of unoccupied molecular orbitals moves downward, while those of occupied molecular orbitals move upward. The redshift of the absorption spectra in **SCPP**s can be ascribed to the slight shift of the frontier molecular orbitals. Moreover, the excitation spectra were measured and compared with the absorption spectra (Fig. 5a–c). The excitation spectra of all **SCPP**s agreed well with the corresponding absorption spectra. For example, according to the excitation spectrum, the high-energy emission at 392 nm of **SCPP[10,8]** was easily detected under excitation at 320 nm, which probably corresponds to the absorption band at 310 nm. The emission at 467 nm can be detected when the excitation wavelength was 340 nm, which is consistent with the absorption peak at 340 nm, and the emission centered at 545 nm arose primarily when the excitation was ~360 nm, which also agrees well with the absorption peak at

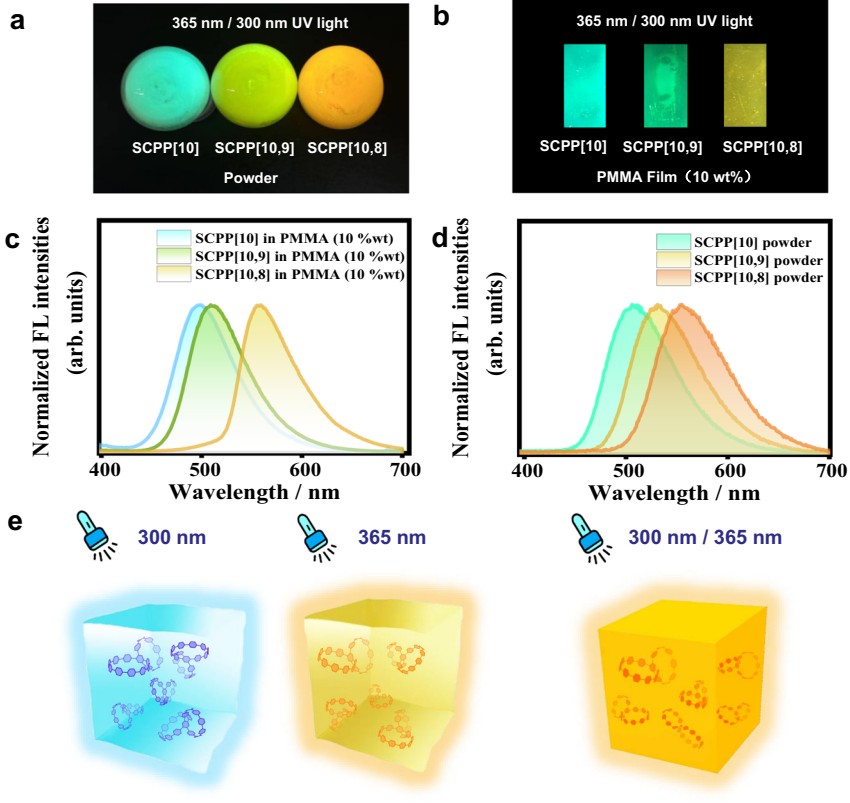

**Fig. 4 | Solid-state emissions of** SCPPs. Fluorescence photographs of **SCPP**s powder (**a**) and PMMA film (**b**) under 300/365 nm UV light. Emission spectra of **SCPP**s powder (**c**) and PMMA film (**d**). **e** Fluorescent behavior of **SCPP[10,8]** in a dilute solution (left and middle) and in solid state (right).

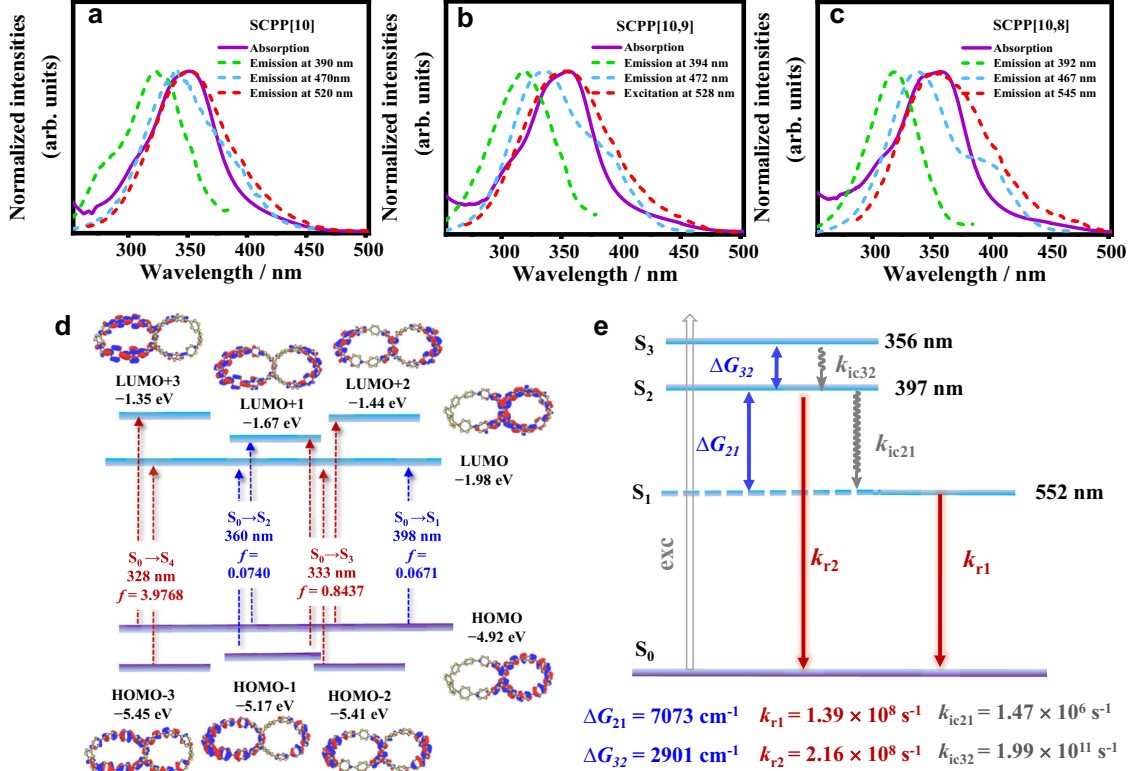

**Fig. 5 | Experimental and computational verification of anti-Kasha emission.**
**a** Excitation spectra of **SCPP[10]** from emissions monitored at 390 nm (green dotted line), at 470 nm, and at 520 nm (blue dotted line); absorption spectrum of **SCPP[10]** (purple solid line). **b** Excitation spectra of **SCPP[10,9]** from emissions monitored at 394 nm (green dotted line), at 472 nm, and at 528 nm (blue dotted line); absorption spectrum of **SCPP[10,9]** (purple solid line). **c** Excitation spectra of

**SCPP[10,8]** from emissions monitored at 392 nm (green dotted line), at 467 nm, and at 545 nm (blue dotted line); absorption spectrum of **SCPP[10,8]** (purple solid line). **d** TDDFT-calculated molecular orbitals and energy diagrams. The $f$ value represents the oscillator strength. **e** Jablonski diagram showing the calculated photophysical processes of **SCPP[10,8]**. $k_{IC}$ internal conversion rate, $k_r$ radiative transition rate, $\Delta G$ energy gap between two excited states.

360 nm. The excitation and absorption spectra of **SCPP[10]** and **SCPP[10,9]** also showed the same characteristics. These data provide strong evidence for emissions derived from both the upper and lowest excited states.

To provide deeper insight into the fluorescence process, excited-state calculations were performed. The calculation results indicated that the allowed transitions of all **SCPP**s are $S_2 \rightarrow S_0$ and $S_1 \rightarrow S_0$ (Fig. 5e and Supplementary Figs. 29–31), showing the anti-Kasha emission feature. The high-energy emission (peak at ~390 nm) of these **SCPP** bisnanohoops comes from the $S_2$ state emission, the lowest energy emission derives from the $S_1$ state emission, and the emission at ~470 nm is probably from the vibronic mixing of the $S_1$ state with higher electronic states. Calculations and experiments have shown that the **SCPP** bisnanohoops have similar $S_2$ excited states with an emission peak at 384–397 nm and $S_3$ excited states with an emission peak 350–357 nm. However, as the size of the bisnanohoop decreases, the higher strain energy leads to a decrease in the torsional angle and increases the degree of conjugation. As a result, the increased conjugation further enhances vibration coupling and reduces $S_1$-$S_0$ energy, resulting in a gradual redshift of the low energy emission.

The validity of Kasha's rule is based on the rates of internal conversion, intersystem crossing, and vibrational relaxation from a given excited state to another excited state being much faster than the intrinsic radiative rate of that state[8]. As mentioned before, prototypical examples with anti-Kasha emission are azulene and thioketones, of which the rates of radiative $S_n \rightarrow S_0$ decay can compete with internal conversion to the $S_1$ state owing to the large energy gap between the $S_2$ and $S_1$ states. Similarly, the excited states of the **SCPP** bisnanohoops were characterized by a large energy separation between the $S_1$ and $S_2$

states, and $\Delta E(S_2 - S_1)$ was calculated to be 6299 cm$^{-1}$ for **SCPP[10]**, 7401 cm$^{-1}$ for **SCPP[10,9]**, and 7073 cm$^{-1}$ for **SCPP[10,8]**. Therefore, $S_2$ fluorescence can compete favorably with internal conversion because of such a large energy gap. The internal conversion rate ($k_{IC}$) and radiative transition rate ($k_r$) were calculated based on different excited states. The $k_r$ of all **SCPP**s is approximately $10^8$ s$^{-1}$, which is faster than the internal conversion rate of $S_2 \rightarrow S_1$ ($k_{IC} = 1.47 \times 10^6$ for **SCPP[10,8]**, $k_{IC} = 2.26 \times 10^5$ s$^{-1}$ for **SCPP[10,9]**, and $k_{IC} = 5.05 \times 10^7$ s$^{-1}$ for **SCPP[10]**). Therefore, under high-energy excitation, **SCPP**s behaved as the anti-Kasha emission of $S_2 \rightarrow S_0$. In addition, the luminescence rate of the $S_1$ state of all **SCPP**s is ~$10^8$ s$^{-1}$, which is in the range of the fluorescence emission rate ($10^7$–$10^{10}$ s$^{-1}$), resulting in $S_1 \rightarrow S_0$ emission under low-energy excitation. All these calculation results are consistent with the excitation-dependent multicolor emission phenomenon of **SCPP** bisnanohoops.

It is worth noting that the $S_1 \rightarrow S_0$ emission of **SCPP**s is significantly different from that of the azulene system. The radiative decay rate of the $S_1$ state of the latter is very low ($k_{r1} = 1.00 \times 10^5$ s$^{-1}$); therefore, it is difficult to observe the emission of the $S_1 \rightarrow S_0$ state[40]. Azulene derivatives were constructed to obtain interesting dual fluorescence by adjusting the energy gaps between the electronic states[41]. As mentioned above, the radiative transition rates of the $S_2$ and $S_1$ excited states for **SCPP**s are both in the range of the fluorescence emission rate, resulting in more abundant and flexible luminescent properties of **SCPP**s than the classic azulene system.

## Emission lifetimes and TRES for SCPP[10,8]

Furthermore, the time-resolved photoluminescence (TRPL) technique was used to study the excited-state lifetimes of **SCPP**s in DCM

(Fig. 6a–c and Supplementary Figs. 32–33). Emission lifetimes of **SCPP**s at ~390 nm were measured upon excitation at 315 nm, and emission lifetimes at ~470 nm, 520 nm, 528 nm and 545 nm were measured upon excitation at 375 nm. All plots can be fitted by biexponential functions, and the fitting results are listed in Supplementary Table 7. **SCPP**s showed a short-lived species and a long-lived species for all monitored emissions, suggesting that two populations are involved in the decay processes. The long $S_2$ state lifetime ($\tau_{S_2} > 10^{-10}$ s) also demonstrated that a large $\Delta E(S_2 - S_1)$ energy gap and a high radiative rate constant ($k_{S_2 \to S_0} \sim 10^8$ s$^{-1}$) are responsible for anomalous fluorescence from the $S_2$ state.

Excited-state decay processes were also studied by time-resolved emission spectroscopy (TRES) (Fig. 6d). The TRES of **SCPP[10,8]** was measured upon excitation at 375 nm. The time-dependent data of **SCPP[10,8]** were collected at wavelengths across the emission band ranging from 378 to 730 nm. The TRES showed three distinct emission spectra, which is consistent with the steady-state emission spectrum. At short delay time (4.85 ns), the spectrum showed a narrow emission peak at 390 nm and a broad emission band maximized at 550 nm with a small shoulder. At a longer delay time, the high-energy emission peak at 390 nm ($S_2$ emission) decreased. The emission intensity of the low-energy emission peak at 550 nm ($S_1$ emission) and the shoulder increased, and two comparable emission bands were detected at 5.31 ns. These observations indicated that the medium emission band (maximized at ~475 nm) and $S_1$ emission are partly from the vibrational relaxation of the $S_2$ excited state. As the delay time increased, the low-energy emission decreased faster, and the high-energy emission decreased relatively slower. In addition, the emission maximized at 475 nm became dominant, while the emission band maximized at 550 nm appeared only as a small shoulder. The results of TRES were matched with emission lifetimes of different excited states.

## Discussion

Overall, we can conclude that all-benzenoid heterogeneous bisnano-hoops demonstrated anomalous excitation-dependent multiple emissions with anti-Kasha characteristics. The spectroscopic results confirmed that the emission processes of bisnanohoops include the participation of both the upper excited states and the lowest excited state. Furthermore, the excited-state calculations indicated that there are large energy gaps between the $S_2$ and $S_1$ excited states in these bisnanohoops, resulting in anti-Kasha emission. Compared with the $S_2$ dominant emission of azulene, our present heterogeneous bisnano-hoops exhibit obvious $S_2$ and $S_1$ dual excited-state emission. Heterogeneous bisnanohoops have the advantages of an interesting symmetry-breaking structure, unique dual excited-state emission, and dynamic adjustability and are attractive organic luminescent materials for future potential applications.

## Methods

### Materials and general information

NMR spectra were recorded on a Bruker BioSpin ([1]H 400 MHz, [13]C 100 MHz) spectrometer or a JEOL JNM-ECZ600R ([1]H 600 MHz, [13]C 150 MHz) NMR spectrometer. Chemical shifts were reported as the delta scale in ppm relative to tetramethylsilane ($\delta = 0.00$ ppm), CDCl$_3$ ($\delta = 7.26$ ppm) for [1]H NMR and CDCl$_3$ ($\delta = 77.0$ ppm) for [13]C NMR. Data were reported as follows: chemical shift, multiplicity (s = singlet, d = doublet, t = triplet, m = multiplet), coupling constant (Hz), and integration. High resolution mass spectrometry (HR-MS) analyses were carried out using MALDI-TOF-MS techniques. All solvents for syntheses were dried by distillation under nitrogen prior to use (tetrahydrofuran and 1,4-dioxane were distilled after reflux with sodium under nitrogen). Other chemicals were obtained from commercial suppliers (Innochem or Acros) and the purity of chemicals are greater than 99%. Air-sensitive reactions were all carried out under argon.

### Synthesis of compound 1

The synthesis of compound **1**[42] was carried out using a modified procedure: To a degassed solution of chlorine substituted precursor (2.8 g, 3.24 mmol) and bis(pinacolato)diboron (3.28 g, 12.94 mmol) in 100 mL of dry 1,4-dioxane was added KOAc (3.18 g, 32.34 mmol) under argon, then the mixture was degassed for 15 min. Pd(OAc)$_2$ (85 mg, 0.38 mmol) and S-Phos (313 mg, 0.76 mmol, 24 mol%) were added to the mixture. The mixture was degassed 10 min, after which it was heated with stirring under argon atmosphere at 90 °C for 48 h. After the reaction mixture was cooled to room temperature, the solvent was removed under reduced pressure and the residue was extracted with dichloromethane. The combined organic layer was washed with brine and dried over anhydrous MgSO$_4$, filtered and concentrated under reduced pressure. Then the resulting product was dissolved in petroleum ether/CH$_2$Cl$_2$ (50:50 mL) and passed through a short silica gel column. After removal of volatiles under reduced pressure to afford a crude product, further purification by an ultrasound wash in cold ethanol to afford pure compound **1** (3 g, 2.85 mmol, 89% yield) as a white solid. [1]H NMR (CDCl$_3$, 400 MHz): $\delta$ (ppm) 7.76 (dd, $J = 7.8$, 6.1 Hz, 4H), 7.55–7.50 (m, 4H), 7.45 (dd, $J = 7.2$, 3.2 Hz, 4H), 7.44–7.38 (m, 4H), 7.36 (s, 4H), 6.11 (dd, $J = 13.2$, 10.3 Hz, 12H), 3.47–3.40 (m, 18H), 1.33 (t, $J = 6.7$ Hz, 24H). [13]C NMR (CDCl$_3$, 100 MHz): $\delta$ (ppm) 146.37, 146.35, 142.73, 142.62, 142.42, 142.33, 139.98, 139.91, 134.87, 134.85, 133.34, 133.32, 133.27, 133.20, 127.08, 126.33, 126.02, 126.00, 125.27, 125.23, 83.70, 74.87, 74.86, 74.62, 74.60, 74.58, 51.95, 51.91, 24.80, 24.78. The characterization matches well with the reported data[42].

### Synthesis of compound 2

The synthesis of compound **2**[43] was carried out using a modified procedure: 1,1'-(1,4-dimethoxy-2,5-cyclohexadiene-1,4-diyl)bis[4-[4-(4-bromophenyl)-1,4-dimethoxy-2,5-cyclohexadien-1-yl]benzene] (1.0 g, 1.14 mmol) was dissolved in THF (15 ml) and cooled to −78 °C. To this solution was added a 2.5 M solution of *n*-BuLi in hexanes (1.0 mL, 2.50 mmol) over 2 min. Then, 2-isopropoxy-4,4,5,5-tetramethyl-1,3,2dioxaborolane (0.6 mL, 2.89 mmol) was added rapidly and the solution was stirred for 20 min. Water (10 mL) was added to the solution and the mixture was allowed to stir for 15 min at room temperature. The aqueous layer was extracted with CH$_2$Cl$_2$ and concentrated under reduced pressure. The crude product was purified by column chromatography (hexane/ethyl acetate = 1/1, *v*/*v*) to afford compound **2** (0.9 g, 85% yield) as a white solid. [1]H NMR (CDCl$_3$, 400 MHz): $\delta$ (ppm)

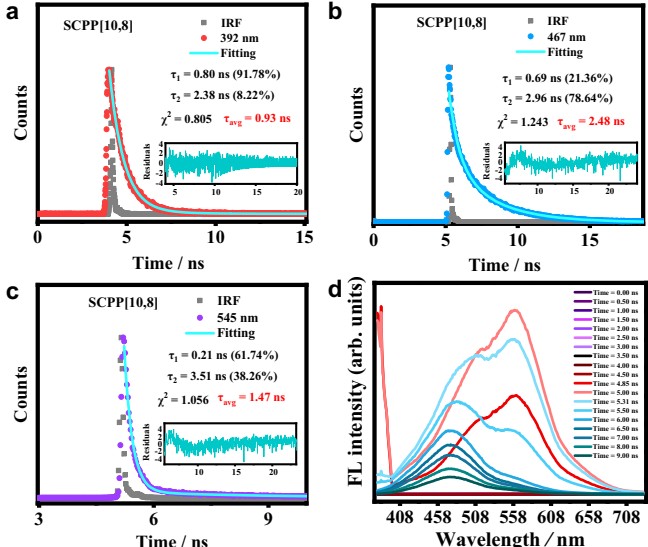

**Fig. 6 | Emission lifetimes and TRES for** SCPP[10,8]**. a–c** Emission lifetimes for **SCPP[10,8]**. **d** Time-resolved emission spectrum for **SCPP[10,8]**.

7.78 (dd, $J$ = 17.6, 8.3 Hz, 4H), 7.58–7.48 (m, 2H), 7.40 (d, $J$ = 8.3 Hz, 4H), 7.34 (s, 6H), 6.09 (dt, $J$ = 5.9, 3.3 Hz, 12H), 3.42 (dd, $J$ = 4.6, 2.6 Hz, 15H), 3.28–3.25 (m, 3H), 1.33 (d, $J$ = 2.5 Hz, 24H). $^{13}$C NMR (CDCl$_3$, 100 MHz): δ (ppm) 146.37, 146.34, 142.65, 142.62, 142.60, 142.57, 142.51, 134.81, 134.78, 133.90, 133.85, 133.64, 133.57, 133.29, 133.19, 133.03, 128.26, 125.91, 125.83, 125.22, 125.17, 83.62, 74.74, 74.49, 74.45, 74.21, 51.84, 51.83, 24.74. The characterization matches well with the reported data[43].

## Synthesis of compound 3

The synthesis of compound **3**[26] was carried out using a modified procedure: To a mixture of benzyloxy substituted [10]cycloparaphenylene (44 mg, 0.045 mmol), 1,2,4,5-tetramethylbenzene (60 mg, 0.45 mmol) under N$_2$ atmosphere was added anhydrous dichloromethane (20 mL). Boron trichloride solution (0.22 ml, 1 M in hexane, 0.22 mmol) was added dropwise to the reaction mixture at −78 °C. The mixture was stirred at −78 °C for 30 min. After warmed up to room temperature, the mixture was quenched with H$_2$O. The organic and aqueous layer was then separated, and the aqueous layer was extracted with CH$_2$Cl$_2$. The organic layers dried with anhydrous Na$_2$SO$_4$ and concentrated under reduced pressure. The crude product was used in the next step without further purification. To the above crude product was added anhydrous dichloromethane (20 mL) under N$_2$ atmosphere. The mixture was added pyridine (0.1 mL, 1.25 mmol) and trifluoromethanesulfonic anhydride (Tf$_2$O, 0.1 mL, 0.60 mmol) at 0 °C. The mixture was then warmed to room temperature and stirred for 12 h. The reaction was quenched with H$_2$O. The organic and aqueous layer was then separated, and the aqueous layer was extracted with CH$_2$Cl$_2$. The organic layers dried with anhydrous Na$_2$SO$_4$ and concentrated under reduced pressure. The residue was purified by column chromatography (hexane/CH$_2$Cl$_2$ = 3/1, $v/v$) to afford compound **3** (30 mg, ~55% over two steps) as a yellow solid. $^1$H NMR (CDCl$_3$, 400 MHz): δ (ppm) 7.65–7.52 (m, 32H), 7.48 (d, $J$ = 8.7 Hz, 4H), 7.35 (s, 2H). $^{13}$C NMR (CDCl$_3$, 100 MHz): δ (ppm) 145.31, 141.26, 138.64, 138.42, 138.21, 138.19, 138.14, 137.96, 137.81, 134.78, 131.98, 129.57, 127.85, 127.47, 127.42, 127.35, 125.43. The characterization matches well with the reported data[26].

## Synthesis of SCPP[10, 9]

To a mixture of **1** (42 mg, 40 μmol), **3** (42 mg, 40 μmol) and K$_2$CO$_3$ (66 mg, 0.48 mmol) in a round-bottomed flask (500 mL) were added THF (150 mL), H$_2$O (30 mL), and Pd(PPh$_3$)$_4$ (10 mg, 9 μmol) under an argon atmosphere. Thereafter, the solution was heated at 75 °C for 48 h. After cooling to room temperature, the solvent was removed under vacuum and the remaining aqueous fraction was extracted with CH$_2$Cl$_2$. The combined organic layer was dried over anhydrous MgSO$_4$ and concentrated under reduced pressure to afford crude product as a yellow solid that was used in the next step without further purification. To a 50-mL round-bottom flask containing a magnetic stirring bar were added SnCl$_2$·2H$_2$O (110 mg, 0.49 mmol), THF (10 mL) and concentrated HCl/H$_2$O (0.1 ml, 12 mol/L) were added, and the resultant mixture was further stirred at room temperature for 30 min. The solution of H$_2$SnCl$_4$/THF was added dropwise to a solution containing the above crude product in 2 mL of THF. After stirring the mixture at room temperature for 2 h, the mixture was quenched with aqueous sodium hydroxide, extracted with CH$_2$Cl$_2$, dried over Na$_2$SO$_4$, and concentrated under reduced pressure. The crude product was purified by column chromatography (hexane/CH$_2$Cl$_2$ = 2/1, $v/v$) to afford **SCPP[10, 9]** (10 mg, ~19% over two steps) as a yellow solid. $^1$H NMR (CDCl$_3$, 600 MHz): δ (ppm) 7.75 (s, 2H), 7.73 (d, $J$ = 3.8 Hz, 4H), 7.71 (s, 4H), 7.61 (dd, $J$ = 8.6, 4.5 Hz, 8H), 7.56 (dd, $J$ = 7.9, 4.7 Hz, 24H), 7.53 (s, 8H), 7.52 (d, $J$ = 8.7 Hz, 4H), 7.45 (d, $J$ = 8.6 Hz, 4H), 7.43 (d, $J$ = 8.0 Hz, 4H), 7.40 (d, $J$ = 8.3 Hz, 4H), 7.36 (d, $J$ = 8.2 Hz, 4H). $^{13}$C NMR (CDCl$_3$, 150 MHz): δ (ppm) 141.90, 141.86, 141.79, 141.66, 141.60, 141.25, 141.14, 141.11, 141.00, 140.96, 140.88, 140.86, 140.73, 140.66, 138.24, 137.67,

132.41, 132.27, 132.16, 131.07, 130.81, 130.63, 130.57, 130.50, 130.44, 130.33, 130.27, 130.14. HR-MS (MALDI-TOF) $m/z$ calcd. for C$_{108}$H$_{70}$ [$M$]$^+$: 1367.5320, found: 1367.5511.

## Synthesis of SCPP[10, 8]

To a mixture of **2** (39 mg, 40 μmol), **3** (42 mg, 40 μmol) and K$_2$CO$_3$ (66 mg, 0.48 mmol) in a round-bottomed flask (500 mL) were added THF (150 mL), H$_2$O (30 mL), and Pd(PPh$_3$)$_4$ (10 mg, 9 μmol) under an argon atmosphere. Thereafter, the solution was heated at 75 °C for 48 h. After cooling to room temperature, the solvent was removed under vacuum and the remaining aqueous fraction was extracted with CH$_2$Cl$_2$. The combined organic layer was dried over anhydrous MgSO$_4$ and concentrated under reduced pressure to afford crude product as a yellow solid that was used in the next step without further purification. To a 50-mL round-bottom flask containing a magnetic stirring bar were added SnCl$_2$·2H$_2$O (110 mg, 0.49 mmol), THF (10 mL) and concentrated HCl/H$_2$O (0.1 ml, 12 mol/L) were added, and the resultant mixture was further stirred at room temperature for 30 min. The solution of H$_2$SnCl$_4$/THF was added dropwise to a solution containing the above crude product in 2 mL of THF. After stirring the mixture at room temperature for 2 h, the mixture was quenched with aqueous sodium hydroxide, extracted with CH$_2$Cl$_2$, dried over Na$_2$SO$_4$, and concentrated under reduced pressure. The crude product was purified by column chromatography (hexane/CH$_2$Cl$_2$ = 2/1, $v/v$) to afford **SCPP[10, 8]** (26 mg, ~51% over two steps) as an orange solid. $^1$H NMR (CDCl$_3$, 600 MHz): δ (ppm) 7.78 (d, $J$ = 8.4 Hz, 4H), 7.74 (d, $J$ = 8.6 Hz, 4H), 7.72 (s, 2H), 7.61 (dd, $J$ = 8.4, 5.8 Hz, 8H), 7.56 (dd, $J$ = 12.7, 4.2 Hz, 20H), 7.52 (s, 8H), 7.51 (d, $J$ = 8.3 Hz, 4H), 7.46 (d, $J$ = 8.5 Hz, 4H), 7.43 (d, $J$ = 8.1 Hz, 4H), 7.40 (d, $J$ = 7.5 Hz, 4H), 7.34 (d, $J$ = 6.5 Hz, 4H). $^{13}$C NMR (CDCl$_3$, 150 MHz): δ (ppm) 141.85, 141.77, 141.62, 141.35, 141.31, 141.22, 141.15, 141.07, 141.00, 140.97, 140.85, 140.73, 140.54, 140.50, 140.16, 137.98, 137.51, 133.10, 132.17, 131.60, 130.95, 130.74, 130.59, 130.51, 130.28, 130.16. HR-MS (MALDI-TOF) $m/z$ calcd. for C$_{102}$H$_{66}$ [$M$]$^+$: 1291.5578, found: 1291.5198.

## Spectroscopy analysis

UV-Vis absorption spectra were performed on a UNIC-3802 spectrophotometer. Steady-state fluorescence spectra were obtained with a Horiba FluoroMax-4 compact spectrofluorometer equipped with an ozone-free xenon lamp optics source. The relative quantum yields were measured using anthracene and quinine sulfate as the references. The time-resolved spectra were recorded using a PL lifetime spectrometer (Edinburgh Instruments, LifeSpec-II) based on a time-correlated single photon counting (TCSPC) technique with excitations of 375 nm and 315 nm picosecond laser. **SCPP**s was measured in DCM with a concentration of $5.0 \times 10^{-6}$ M.

## Computational details

Theoretical calculations were performed using the Gaussian 16 software package[39]. All calculations were carried out using the density functional theory (DFT) method. Geometrical structure optimization of three compounds were optimized at the B3LYP/6-31 G(d) level, and the energies were calculated at the CAM-B3LYP/6-31 G(d) level. The strain energy was calculated using the reported computational methods[33,44]. The molecular orbitals were generated by the Multiwfn software (A Multifunctional Wavefunction Analyzer Version 3.8 dev)[45]. The photophysical properties were calculated by using the photophysics module of the Molecular Materials Property Prediction Package (MOMAP Version 2021 A 2.3.1)[46–52]. The Gaussian 16 software package was used to perform geometry optimization and frequency calculations on the initial and final states. The bi-exponential decays indicate that there are two possible populations with different radiative lifetimes involved in the S$_2$ and S$_1$ emissions. These two transitions probably originate from the different configurations of the molecule in the excited state, with one configuration being the dominant

component. Due to the complexity of excited state configuration, it is difficult to determine which specific configurations are involved in the luminescence process. In order to simplify the calculation process, the minimum energy geometry in the excited state was chosen as the only configuration for the calculation of the radiation rate and internal conversion rate. The transition dipole moment and the transition electric field between the two states were calculated and used for the $k_r$ calculation. The non-adiabatic coupling matrix element (NACME) was obtained by Gaussian 16 for the $k_{IC}$ calculation.

## Data availability

Characterization studies, $^1H$ NMR spectra, $^{13}C$ NMR spectra, mass spectrometry data and useful information are available in the Supplementary Information. Coordinates of the optimized structures are provided in the source data file. Additional data that support the findings of this study are available from the corresponding author upon request. Source data are provided with this paper.

## Code availability

The Molecular Material Property Prediction Package (MOMAP) workshop is commercially available at http://www.momap.net.cn/. The Multifunctional Wavefunction Analyzer (Multiwfn) program is free access at http://sobereva.com/multiwfn.

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

## Acknowledgements
This work was financially supported by the National Natural Science Foundation of China (22225108, P.D.; 21971229, P.D.), and Hefei National Research Center for Physical Sciences at the Microscale. This work was partially carried out at the Instruments Center for Physical Science, University of Science and Technology of China.

## Author contributions
P.D. conceived and designed this research. X.Z. synthesized all the compounds, conducted all characterizations, and physical measurements. C.C., G.Z. and X.W. did all the calculation studies. X.Z., W.Z., N.Y., B.Y., G.Z., X.W. and P.D. co-wrote the paper, and all the authors commented on it.

## Competing interests
The authors declare no competing interests.
