## [Peer Review File · Nature Communications]

REVIEWER COMMENTS

Reviewer #1 (Remarks to the Author):

Zhang et al report a series of heterogeneous carbon nanohoops that exhibit interesting emission behaviour. With the excitation energy dependence and calculations the authors claim that both Kasha (S₀-S₁) emission and anti-Kasha (S₀-S₂) emission is present in these systems. The work is interesting and well presented, although a number of clarifications must be added before the work can be fully assessed, see below. I thus recommend major revision of the manuscript.

1. For a complete photophysical presentation the absorption spectra of the SCPPs could be included earlier in the main text, together with the fluorescence spectra.
2. Fig 3 a-c, the fluorescence spectra at 365 nm excitation is visually blocking the fluorescence spectra at 300 nm excitation, hence it is not possible to see how the two spectra overlap. I suggest using a semitransparent fill so that the bottom spectra is visible for comparison.
3. Figure 4e, what are the authors trying to illustrate. In the text the authors claim there is excitation wavelength dependence on the emission in solid state, however my interpretation of the illustration in 4e is that at 300 nm high energy blue light is observed, 365 nm excitation results in yellow emission and the combined emission a stronger yellow is observed, hence this illustration does describe an excitation wavelength dependence. Can the authors explain and clarify this?
4. The discussion on the solid state emission is not clear to me. The authors claim that “emissions from upper excited states other than the lowest singlet are very difficult to detect in condensed phases due to significant amounts of collisional relaxation or fast vibrations of upper excited states”, I disagree with the formulation, e.g. in the solid state collisions would be decreased as compared to in liquid solutions. Yet intermolecular interactions can still lead to increased deactivation pathways. For example, in dilute PMMA films, as used by the authors, collisions are minimized compared to solution and often vibrational relaxation is decreased in a rigid matrix such as PMMA. With this weak argumentation the conclusion in the section is not supported “Therefore, the single excitation-independent emission of SCPPs in the solid-state also implies the possibility of upper excited state emissions, which typically occurs under collision-free conditions or in very dilute solutions.” Further more, later in the text the authors assign the ~550 nm emission to S₀-S₁ emission, which is the emission observed here in the solid state, hence it seems no anti-Kasha behaviour or involvement of upper excited states are present in the solid state.

5. The authors should specify how the rates were calculated and not just state their values. Since the time-resolved photoluminescence yielded multi-exponential decays it is not straight forward to calculate these rates and a clear description and motivation to assumptions should be included.

6. In a couple of places the authors state that “while the

long-lived emission at 475 nm is probably from a vibrational energy level”, I assume that they mean an vibrational energy level of the S1 state? In which case it is not clear why this emission should be longer lived than the emission from S1 (or the S2) state. The authors must clarify this and support the designation of this emission.

7. I am also curious what are the quantum yields of emission for the different emission processes?

Reviewer #2 (Remarks to the Author):

Du and Co workers have submitted a manuscript to be considered for publication in Nature Communications. The manuscript describes their work on bis(nanohoops), of which they have synthesised 3 novel examples. The synthesis relies on methodology which is by now well established involving formation of a macrocyclic precursor bearing sp³ centres (and hence which is of comparatively low strain), followed by the reductive aromatisation of these compounds to afford the desired nanohoops, in which the increase in strain energy is offset by the increase in aromaticity. As such the synthetic part of the work is not particularly impactful. Rather, the significance of the work described in this manuscript lies in the photophysical properties of these bis(nanohoops). The authors have comprehensively characterised these novel molecules using various photophysical techniques, and have shown convincingly that they violate Kasha's rule, by exhibiting significant emission from an excited state higher than S₁. Genuine examples of molecules which violate Kasha's rule are very few and far between, and as the authors correctly point out their work constitutes the first example of such systems that comprise solely benzenoid aromatic rings. The spectroscopic characterization is backed up by a comprehensive computational study which serves to elucidate the underlying processes that give rise to the dual fluorescence that these molecules show. If these molecules were entirely devoid of potential applications, I would nevertheless consider this work to be sufficiently significant to merit publication in this journal. However, this is not the case, and the authors are justified in claiming that such tunable dual emitters will find application in multiple areas. In support of this contention the authors may wish to include a few supporting references, for example NATURE COMMUNICATIONS, 2020, 11, 793. The potential uses to which these novel molecules (or derivatives thereof) could be put serves to further increase the significance of this work. As such I believe the necessary criteria to merit publication in this journal have been reached, and I am happy to recommend that the manuscript be accepted for publication, subject only to some minor changes, listed below.

- Introduction: the authors should add the following reference when describing background literature on CPPs: Chem. Soc. Rev., 2015, 44, 2221
- Figure 1: The font “Comic Sans” has been used in this figure. This font is often used for “informal” documents. For a scientific paper I recommend using a standard font like “Arial”, “Helvetica” or “Times New Roman”, like the authors have used in Figure 2
- Figure 1b: “Thiophos” should be “Thiophosgene”. Actually, “Thiophos” is an insecticide! (diethyl paranitrophenyl thiophosphate). Also the name “xanthione” should be included in Figure 1b
- Figure 5a,b,c: The legend for the different colours of dashed lines say “Excitation at 390 nm, Excitation at 470 nm, Excitation at 520nm”. But actually they should say “Emission at 390 nm, Emission at 470 nm, Emission at 520nm”, etc...
- “It is worth noting that the S1 → S0 emission of SCPPs is significantly different from that of the azulene system.” in this paragraph the authors should include a reference to the following computational study which determined that the S1 excited state of azulene decays via a conical intersection: J. Am. Chem. Soc. 1996, 118, 169. The authors should also mention the fact that certain substituted azulenes which have a somewhat smaller S2-S1 energy gap are known to exhibit dual fluorescence: Chem. Phys. Lett. 1974, 29, 397.

Reviewer #3 (Remarks to the Author):

Authors report new bisnanohoops and anti-Kasha luminescence, which is well-supported by excitation, absorption, and emission spectra as well as DFT calculations. The photophysical properties are interesting, and given the rarity of true anti-Kasha emission, the manuscript can possibly be made suitable for publication in this journal. Specific comments below:

The yield of SCPP[10,8] is missing.

The authors state that successful synthesis is supported by techniques, including “various NMR spectra”. While COSY and HSQC spectra are provided in the SI, I find the 2D resolution to be quite poor, and the authors have not provided any interpretation or assignments of chemical shifts. Therefore, beyond integration of ¹H resonances and counting of ¹³C peaks, the various extra NMR spectra do not help convince the reader that these complex structures have been prepared.

I suggest moving the paragraph “Theoretical Calculations of SCCPs” to after the Anti-Kasha emission spectra. I can see why the authors located the geometric discussion upfront, but it is more related to an explanation of the results than a characteristic of the molecules. It would be more exciting to lead the results section with the spectroscopy.

Biexponential fits to the emission lifetime data show that two components are present in ratios of ~80:20. It would be useful to show the expansion of the data, the biexponential fits, and the residuals. In addition, the 520 nm data are questionable due to the fast decay and similarity to the instrument response. A biexponential fit may not be sufficient. The authors may need to use a deconvolution of a Gaussian + a biexponential to accurately measure their data.

Point-by-point response to reviewer's comments:

Reviewer #1 (Remarks to the Author):

Zhang et al report a series of heterogeneous carbon nanohoops that exhibit interesting emission behaviour. With the excitation energy dependence and calculations the authors claim that both Kasha (S_0 - S_1) emission and anti-Kasha (S_0 - S_2) emission is present in these systems. The work is interesting and well presented, although a number of clarifications must be added before the work can be fully assessed, see below. I thus recommend major revision of the manuscript.

Response: We thank this reviewer very much for these positive comments and helpful suggestions to our work.

1. For a complete photophysical presentation the absorption spectra of the SCPPs could be included earlier in the main text, together with the fluorescence spectra.

Response: We appreciate the reviewer's good comment to improve our manuscript. Following the reviewer's suggestion, we have added the absorption and fluorescence spectra in the main context (see following Figure 3a-3c).

Fig. 3 Anti-Kasha emissions of SCPPs in solution. a-c Absorption and emission spectra (under excitation at 360 nm) of SCPPs. d-e Fluorescence photographs and emission spectra of SCPPs under 300/365 nm UV light in dilute DCM ($c = 5 \times 10^{-6}$ mol/L). h-g Excitation-dependent emissive properties of SCPPs.

The related description (page 7, in the first paragraph) has been revised as follows:

“...As shown in Fig. 3a-3c, the UV-Vis absorption spectra of SCPPs exhibited broad absorption bands in the region of 280 - 480 nm. The maximum absorption peaks appear at 359 nm, 356 nm, and 351 nm for SCPP[10,8]-SCPP[10], respectively. In addition,

there are an obvious shoulder peak at approximately 340 nm and a weak shoulder peak in the longer wavelength region (at ~ 400 nm) for all SCPPs.

Furthermore, the steady-state photoluminescence (PL) spectra of SCPP[10], SCPP[10,9], and SCPP[10,8] were preliminarily measured under excitation at 360 nm (Figs. 3a-3c). Each SCPP has a size-dependent low-energy emission (520 nm for SCPP[10], 528 nm for SCPP[10,9], and 545 nm for SCPP[10,8]) and two high-energy shoulder peaks at around 470 nm and 390 nm.”

2. Fig 3 a-c, the fluorescence spectra at 365 nm excitation is visually blocking the fluorescence spectra at 300 nm excitation, hence it is not possible to see how the two spectra overlap. I suggest using a semitransparent fill so that the bottom spectra is visible for comparison.

Response: We thank this reviewer very much for pointing out this. We have corrected and used the semitransparent fill for a clearer view and comparison, as this reviewer suggested in the revised version (see following Figure 3d-f).

Fig. 3 Anti-Kasha emissions of SCPPs in solution. a-c Absorption and emission spectra (under excitation at 360 nm) of SCPPs. d-e Fluorescence photographs and emission spectra of SCPPs under 300/365 nm UV light in dilute DCM ($c = 5 \times 10^{-6}$ mol/L). h-g Excitation-dependent emissive properties of SCPPs.

3. Figure 4e, what are the authors trying to illustrate. In the text the authors claim there is excitation wavelength dependence on the emission in solid state, however my interpretation of the illustration in 4e is that at 300 nm high energy blue light is observed, 365 nm excitation results in yellow emission and the combined emission a stronger yellow is observed, hence this illustration does describe an excitation wavelength dependence. Can the authors explain and clarify this?

Response: Thanks for this point. The purpose of the schematic diagram Figure 4e is to summarize the fluorescence characteristics of **SCPP[10,8]** in different states under different irradiation wavelengths. In a dilute solution, **SCPP[10,8]** showed blue fluorescence under irradiation at 300 nm UV light and orange fluorescence under irradiation at 365 nm UV light, which exhibits excitation-dependent emission with anti-Kasha behavior. However, in the solid state, **SCPP[10,8]** exhibited a single and bright fluorescence, and its emission did not change with different excitations (Kasha emission). We have corrected the figure legend of Figure 4e, which has been modified as follows (see following Figure 4e):

Fig. 4 Solid-state emissions of SCPPs. Fluorescence photographs of SCPPs powder **a** and PMMA film **b** under 300/365 nm UV light. Emission spectra of SCPPs powder **c** and PMMA film **d**. **e** Fluorescent behavior of **SCPP[10,8]** in a dilute solution (left and middle) and in solid state (right).

4. The discussion on the solid state emission is not clear to me. The authors claim that “emissions from upper excited states other than the lowest singlet are very difficult to detect in condensed phases due to significant amounts of collisional relaxation or fast vibrations of upper excited states”, i disagree with the formulation, e.g. in the solid state collisions would be decreased as compared to in liquid solutions. Yet intermolecular interactions can still lead to increased deactivation pathways. For example, in dilute PMMA films, as used by the authors, collisions are minimized compared to solution and often vibrational relaxation is decreased in a rigid matrix such as PMMA. With this weak argumentation the conclusion in the section is not supported “Therefore, the single excitation-independent emission of SCPPs in the solid-state also implies the possibility of upper excited state emissions, which typically occurs under collision-free conditions or in very dilute solutions.” Furthermore, later in the text the authors assign

the ~550 nm emission to S₀-S₁ emission, which is the emission observed here in the solid state, hence it seems no anti-Kasha behaviour or involvement of upper excited states are present in the solid state.

Response: We thank this reviewer very much for this comment and we agree with this reviewer. The fluorescence behavior of molecules in different states is affected by many factors. In dilute solutions, the fluorescence emission of molecules can be affected by various non-radiative excited state energy loss pathways such as vibrational relaxation and intramolecular rotations. Collision with solvent molecules is another important factor by which the excited state energy is lost non-radiatively. In a rigid matrix or aggregate state, as stated by this reviewer, the collisions are minimized compared to the solution and often vibrational relaxation is decreased. Besides, various intermolecular interactions, such as complex formation, intense intermolecular π - π stacking, and energy transfer, will also influence the fluorescence emission. Therefore, as stated, the argumentation that “Therefore, the single excitation-independent emission of SCPPs in the solid-state also implies the possibility of upper excited state emissions...” is weak and unpersuasive, and we have deleted this conclusion in the revised version. Besides, we assigned the 545 nm emission of SCPP[10,8] to S₁-S₀ emission, and this lowest excited state emission can be clearly observed in both solution and solid state. We also agree with this reviewer that SCPP[10,8] no anti-Kasha behavior or involvement of upper excited states are present in the solid state, and this point was also described in our manuscript.

The related description (*page 9, in the first paragraph*) has been revised as follows: “The solid-state fluorescence properties of SCPPs have also been studied. As observed with the naked eye, the color of SCPPs gradually changed from light yellow to orange as the ring size decreased (Supplementary Fig. 15). All these solid samples exhibited a single and bright fluorescence under a handheld UV light, and their emissions did not change with different excitation wavelengths (Figs. 4a, 4c and Supplementary Fig. 16). To further verify this phenomenon, the PL spectra of SCPP films were measured, and the results confirmed the independence of excitation (Figs. 4b, 4d and Supplementary Fig. 17). Therefore, SCPPs exhibit excitation-dependent multiple emission properties in solution, which violates the Kasha’s rule. However, in the solid state, the high energy emissions vanished and only showed the lowest energy emissions, which obeys the Kasha’s rule (Fig. 4e).”

5. The authors should specify how the rates were calculated and not just state their values. Since the time-resolved photoluminescence yielded multi-exponential decays it is not straight forward to calculate these rates and a clear description and motivation to assumptions should be included.

Response: Thank the reviewer for pointing out this. The bi-exponential decays indicate that there are two possible transitions involved in the S₂ and S₁ emission, and (taking SCPP[10,8] as an example) the ratio of these two transitions are about 92:8 (at 392 nm), 79:21 (at 467 nm), and 62:38(at 545 nm) (see Figs. 6a-6c, Supplementary Figs. 26-27,

and Supplementary Table 7). These two transitions probably originate from the different configurations of the molecule in the excited state, with one configuration being the dominant component. In order to simplify the calculation process, the minimum energy geometry in the excited state was chosen as the only configuration for the calculation of the radiation rate and non-radiation rate.

The related description (*page 21, in the first paragraph*) has been revised as follows: “...The Gaussian 16 software package was used to perform geometry optimization and frequency calculations on the initial and final states. The bi-exponential decays indicate that there are two possible transitions involved in the S₂ and S₁ emission. These two transitions probably originate from the different configurations of the molecule in the excited state, with one configuration being the dominant component. Due to the complexity of excited state configuration, it is difficult to determine which specific configurations are involved in the luminescence process. In order to simplify the calculation process, the minimum energy geometry in the excited state was chosen as the only configuration for the calculation of the radiation rate and internal conversion rate. The transition dipole moment and the transition electric field between the two states were calculated and used for the k_r calculation. The non-adiabatic coupling matrix element (NACME) was obtained by Gaussian 16 for the k_{IC} calculation.”

6. In a couple of places the authors state that “while the long-lived emission at 475 nm is probably from a vibrational energy level”, I assume that they mean an vibrational energy level of the S₁ state? In which case it is not clear why this emission should be longer lived than the emission from S₁ (or the S₂) state. The authors must clarify this and support the designation of this emission.

Response: As described in the introduction of the manuscript, [n]Cycloparaphenylenes (CPPs) have interesting and counterintuitive optical properties. Due to the conservation of orbital symmetry for the centrosymmetric CPPs, the HOMO-LUMO optical transitions are Laporte forbidden. The S₁ → S₀ transition is symmetry-forbidden, therefore why CPP can emit high-efficiency fluorescence is a question worth exploring. According to reports of Irlé (*Chem. Sci.*, **2013**, 4, 187; *J. Chem. Theory Comput.* **2014**, 10, 4025–4036), the efficient photoluminescence of CPPs comes from the vibronic mixing of the S₁ state with higher electronic states. The formally dipole-forbidden transition between the S₀ and S₁ states is increasingly allowed with increasing ring size owing to the vibronic coupling of the S₁ state with higher excited states through non-totally symmetric vibrational modes. These findings reveal that vibronic effects are important in determining the photophysical properties of CPP. Compared with CPP, SCPPs have symmetry-breaking structure, which allow the HOMO-LUMO transition and S₁ → S₀ transition. The excited-state calculations results indicated that the high-energy emission (peak at ~390 nm) of these SCPP bisnanohoops comes from the S₂ → S₀ emission, and the lowest energy emission derives from the S₁ → S₀ emission. However, it is difficult to assign the emission from 470 nm, and this will require intensive and complex experimental investigation and theoretical calculations. Based

on above analysis, we tentatively explain that the emission at 470 nm probably comes from the vibronic mixing of the S₁ state with higher electronic states.

Thank this reviewer very much for pointing out the problem of long-lived emission at 475 nm. Actually, “long-lived emission” is often used to describe delayed fluorescence or phosphorescence. Besides, the lifetime of emission at 475 nm (2.48 ns) is not significantly different from S₂ (0.93 ns) and S₁ emission (1.47 ns). Therefore, the term “long-lived” is not accurate to describe the emission at 475 nm with just lifetime 2.48 ns. Because the emission at 470 nm may not be a simple vibration of the S₁ state or S₂ state, it perhaps has a different lifetime than the S₂ and S₁ states. We have deleted the inaccurate description of “long-lived emission” and corrected the corresponding description in our manuscript.

The related description (*page 13, in the first paragraph*) has been revised as follows: “...The high-energy emission (peak at ~390 nm) of these **SCPP** bisnanohoops comes from the S₂ state emission, the lowest energy emission derives from the S₁ state emission, and the emission at ~470 nm is probably from the vibronic mixing of the S₁ state with higher electronic states...”

7. I am also curious what are the quantum yields of emission for the different emission processes?

Response: We thank this reviewer very much for this good point. Following the reviewer’s suggestion, we have measured the quantum yields for the different emission processes of **SCPPs**.

Sample	Φ_{S_2} [%]	Φ_{S_1} [%]
SCPP[10]	0.30	3.1
SCPP[10,9]	0.12	1.94
SCPP[10,8]	0.57	1.20

Supplementary Table 1. The quantum yields of emissions for the different emission processes of **SCPPs**.

The corresponding text describing quantum yields (*page 8, in the first paragraph*) has been added as follows: “Besides, fluorescence quantum yields for the different emission processes were measured in dilute solution at room temperature (Supplementary Table 1). The quantum yields from the low energy emission of **SCPPs** decreased as bisnanohoops got smaller, which is similar to the size-dependent emission properties of **CPPs**.”

Overall, we appreciate the reviewers very much for these comments and suggestions to help us to improve our manuscript.

Reviewer #2 (Remarks to the Author):

Du and Co-workers have submitted a manuscript to be considered for publication in Nature Communications. The manuscript describes their work on bis(nanohoops), of which they have synthesised 3 novel examples. The synthesis relies on methodology which is by now well established involving formation of a macrocyclic precursor bearing sp³ centres (and hence which is of comparatively low strain), followed by the reductive aromatisation of these compounds to afford the desired nano hoops, in which the increase in strain energy is offset by the increase in aromaticity. As such the synthetic part of the work is not particularly impactful. Rather, the significance of the work described in this manuscript lies in the photo physical properties of these bis(nanohoops). The authors have comprehensively characterised these novel molecules using various photophysical techniques, and have showed convincingly that they violate Kasha's rule, by exhibiting significant emission from an excited state higher than S₁. Genuine examples of molecules which violate Kasha's rule are very few and far between, and as the authors correctly point out their work constitutes the first example of such systems that comprise solely benzenoid aromatic rings. The spectroscopic characterization is backed up by a comprehensive computational study which serves to elucidate the underlying processes that give rise to the dual fluorescence that these molecules show. If these molecules were entirely devoid of potential applications, I would nevertheless consider this work to be sufficiently significant to merit publication in this journal. However, this is not the case, and the authors are justified in claiming that such tunable dual emitters will find application in multiple areas. In support of this contention the authors may wish to include a few supporting references, for example NATURE COMMUNICATIONS, 2020, 11, 793. The potential uses to which these novel molecules (or derivatives thereof) could be put serves to further increase the significance of this work. As such I believe the necessary criteria to merit publication in this journal have been reached, and I am happy to recommend that the manuscript be accepted for publication, subject only to some minor changes, listed below.

Response: We thank this reviewer very much for the positive comments and great support to our work. As suggested, we have added the reference to the application of dual-emission anti-Kasha-active fluorophores (*Nat. Commun.* **2020**, 11, 793) in the corresponding paragraph (see ref. 14).

- Introduction: the authors should add the following reference when describing background literature on CPPs: *Chem. Soc. Rev.*, 2015, 44, 2221

Response: Thanks. According to the suggestion, we have cited this comprehensive review of cycloparaphenylenes and related nanohoops from Lewis (*Chem. Soc. Rev.*, **2015**, 44, 2221) to describe the background of CPPs (see ref. 35).

- Figure 1: The font "Comic Sans" has been used in this figure. This font is often used

for “informal” documents. for a scientific paper I recommend using a standard font like “Arial”, “Helvetica” or “Times New Roman”, like the authors have used in Figure 2.

Response: We thank the reviewer very much for pointing out the font problem. According to the suggestion, we have changed the font in Fig. 1 to “Times New Roman” (see following Fig. 1).

Fig. 1 Schematic representation. **a** Structure of azulene. **b** Structures of thiophosgene (left) and xanthione (right). **c** Structures of heterogeneous bisnanohoops SCPP[10,9] (left) and SCPP[10,8] (right). **d** Anti-Kasha emission of azulene. **e** S_2 , S_1 dual excited-state emissions of heterogeneous bisnanohoops.

- Figure 1b: “Thiophos” should be “Thiophosgene”. Actually, “Thiophos” is an insecticide! (diethyl paranitrophenyl thiophosphate). Also the name “xanthione” should be included in Figure 1b.

Response: Thanks. As suggested, we have changed the “Thiophos” to “Thiophosgene” and added the name “xanthione” in Fig. 1b (see following Fig. 1b).

Fig. 1 Schematic representation. **a** Structure of azulene. **b** Structures of thiophosgene (left) and xanthione (right). **c** Structures of heterogeneous bisnanohoops **SCPP[10,9]** (left) and **SCPP[10,8]** (right). **d** Anti-Kasha emission of azulene. **e** S_2 , S_1 dual excited-state emissions of heterogeneous bisnanohoops.

• Figure 5a,b,c: The legend for the different colours of dashed lines say “Excitation at 390 nm, Excitation at 470 nm, Excitation at 520nm”. But actually they should say “Emission at 390 nm, Emission at 470 nm, Emission at 520nm”, etc...

Response: We thank this reviewer very much for these good comments. We have corrected the legend as this reviewer suggested (see following Fig. 5a-5c).

Fig. 5 Experimental and computational verification of anti-kasha emission. **a** Excitation spectra of **SCPP[10]** from emissions monitored at 390 nm (green dotted line), at 470 nm, and at 520 nm (blue dotted line); absorption spectra of **SCPP[10]** (purple solid line). **b** Excitation spectra of **SCPP[10,9]** from emissions monitored at 394 nm (green dotted line), at 472 nm, and at 528 nm (blue dotted lines); absorption spectra of **SCPP[10,9]** (purple solid line). **c** Excitation spectra of **SCPP[10,8]** from emissions monitored at 392 nm (green dotted line), at 467 nm, and at 545 nm (blue dotted lines); absorption spectra of **SCPP[10,8]** (purple solid line). **d** TDDFT-calculated molecular orbitals and energy diagrams. The f value represents the oscillator strength. **e** Jablonski diagram showing the calculated photophysical processes of **SCPP[10,8]**. k_{IC} : internal conversion rate, k_r : radiative transition rate, ΔG : energy gap between two excited states.

• “It is worth noting that the $S_1 \rightarrow S_0$ emission of SCPPs is significantly different from that of the azulene system.” in this paragraph the authors should include a reference to the following computational study which determined that the S_1 excited state of azulene

decays via a conical intersection: *J. Am. Chem. Soc.* 1996, 118, 169. The authors should also mention the fact that certain substituted azulenes which have a somewhat smaller S₂-S₁ energy gap are known to exhibit dual fluorescence: *Chem. Phys. Lett.* 1974, 29, 397.

Response: Thank this reviewer for this good comment to improve our manuscript. As suggested, we have cited the computational studies on the decay of the S₁ excited state of azulene (*J. Am. Chem. Soc.* 1996, 118, 169-175, see ref. 40) in the corresponding paragraph. Moreover, we have added statements and related references on azulene derivatives with dual fluorescence (*Chem. Phys. Lett.* 1974, 29, 397-404, see ref. 41). The related description (*page 14, in the second paragraph*) has been revised as follows: “It is worth noting that the S₁ → S₀ emission of **SCPPs** is significantly different from that of the azulene system. The radiative decay rate of the S₁ state of the latter is very low ($k_{r1} = 1.00 \times 10^5 \text{ s}^{-1}$); therefore, it is difficult to observe the emission of the S₁ → S₀ state⁴⁰. Azulene derivatives were constructed to obtain interesting dual fluorescence by adjusting the energy gaps between the electronic states⁴¹...”

Overall, we appreciate the reviewers very much for these comments and suggestions to help us to improve our manuscript.

Reviewer #3 (Remarks to the Author):

Authors report new bisnanohoops and anti-Kasha luminescence, which is well-supported by excitation, absorption, and emission spectra as well as DFT calculations. The photophysical properties are interesting, and given the rarity of true anti-Kasha emission, the manuscript can possibly be made suitable for publication in this journal. Specific comments below:

The yield of SCPP[10,8] is missing.

Response: Thank this reviewer very much for pointing out this. We have added the yield of SCPP[10, 8] in the revised version.

The related description (*page 20, in the first paragraph*) has been revised as follows:

“The crude product was purified by column chromatography (hexane/CH₂Cl₂ = 2/1, v/v) to afford SCPP[10, 8] (26 mg, ~51% over two steps) as an orange solid.”

The authors state that successful synthesis is supported by techniques, including “various NMR spectra”. While COSY and HSQC spectra are provided in the SI, I find the 2D resolution to be quite poor, and the authors have not provided any interpretation or assignments of chemical shifts. Therefore, beyond integration of ¹H resonances and counting of ¹³C peaks, the various extra NMR spectra do not help convince the reader that these complex structures have been prepared.

Response: Thanks. In order to improve spectral resolution, we re-characterized SCPP[10,8] and SCPP[10,9] using a JEOL JNM-ECZ600R (¹H 600 MHz, ¹³C 150 MHz) NMR spectrometer (¹H NMR, ¹³C NMR, 2D ¹H-¹H COSY NMR, 2D NOESY NMR, 2D (H, C)-HSQC NMR, and 2D (H, C)-HMBC NMR) (see following Supplementary Figs. 1-12). Due to the specific all-phenylene structure of SCPPs, the chemical environments of most C and H are very similar, resulting in severe overlap of signals in the NMR spectra. Therefore, only certain chemical shifts in the NMR spectra at specific positions can be resolved. We have assigned these specific signals observed in the NMR spectra of SCPP[10,8] and SCPP[10,9]. In the NMR spectra of SCPP[10,8], the characteristic singlet at 7.72 ppm in ¹H NMR can be assigned to the protons in the bridging benzene unit (H_a). The two doublets at 7.74 and 7.78 ppm are tentatively assigned to H_b and H_d. To fully assign the aromatic proton signals of SCPP[10,8], 2D ¹H-¹H COSY NMR and NOESY NMR spectra were measured. The doublet at 7.74 ppm is correlated with the proton signal at 7.34 ppm, suggesting that these signals can be assigned to H_c. The doublet at 7.78 ppm is correlated with the proton signal at 7.43 ppm, suggesting that these signals can be assigned to H_e. Finally, the remaining signals can be assigned to other protons in the para-linked benzene scaffold. To assign the aromatic carbon signals of SCPP[10,8], 2D (H, C)-HSQC NMR and 2D (H, C)-HMBC NMR spectra were measured. Since singlet H_a is correlated with the carbon signal at 133.10 ppm, it can be assigned to C₁. The H_b and H_d are correlated with the carbon signals at 137.98 and 137.51 ppm, which can be assigned to C₃ and C₂

respectively. The chemical shifts of **SCPP[10,9]** were assigned using the same method. Therefore, various extra NMR spectra and high-resolution mass spectrometry can prove the successful synthesis of heterogeneous bisnanohoops **SCPP[10,8]** and **SCPP[10,9]**.

Supplementary Figure 1. ^1H NMR spectrum of **SCPP[10,8]** (600 MHz, CDCl_3).

Supplementary Figure 2. ^{13}C NMR spectrum of SCPP[10,8] (150 MHz, CDCl_3).

Supplementary Figure 3. Expanded 2D ^1H - ^1H COSY NMR spectrum (600 MHz, CDCl_3) of SPP[10,8].

Supplementary Figure 4. Expanded 2D ^1H - ^1H NOESY NMR spectrum (600 MHz, CDCl_3) of SPP[10,8].

Supplementary Figure 5. Expanded 2D (H, C)-HSQC NMR spectrum (600 MHz, CDCl_3) of SCPP[10,8].

Supplementary Figure 6. Expanded 2D (H, C)-HMBC NMR spectrum (600 MHz, CDCl_3) of SCPP[10,8].

Supplementary Figure 7. ^1H NMR spectrum of SCPP[10,9] (600 MHz, CDCl_3).

Supplementary Figure 8. ^{13}C NMR spectrum of SCPP[10,9] (150 MHz, CDCl_3).

Supplementary Figure 9. Expanded 2D ^1H - ^1H COSY NMR spectrum (600 MHz, CDCl_3) of SCPP[10,9].

Supplementary Figure 10. Expanded 2D ^1H - ^1H NOESY NMR spectrum (600 MHz, CDCl_3) of SCPP[10,9].

Supplementary Figure 11. Expanded 2D (H, C)-HSQC NMR spectrum (600 MHz, CDCl₃) of SCPP[10,9].

Supplementary Figure 12. Expanded 2D (H, C)-HMBC NMR spectrum (600 MHz, CDCl₃) of SCPP[10,9].

I suggest moving the paragraph “Theoretical Calculations of SCCPs” to after the Anti-Kasha emission spectra. I can see why the authors located the geometric discussion upfront, but it is more related to an explanation of the results than a characteristic of the molecules. It would be more exciting to lead the results section with the spectroscopy.

Response: Thank this reviewer for this good comment to improve our manuscript. Following the reviewer’s suggestion, we have moved the paragraph “Theoretical Calculations of SCCPs” after the anti-Kasha emission spectra. The description and comparison of absorption and emission spectra were used to elicit unusual anti-Kasha properties, which may make this manuscript more exciting.

The related description (*page 6, in the first paragraph*) has been revised as follows:

“**Anti-Kasha emissions of SCCPs in solution.** To investigate the regular characteristics of a series of bisnanohoops, the homogeneous bisnanohoop **SCPP[10]** was also studied for comparison²⁶. The photophysical properties of **SCPP[10,8]**, **SCPP[10,9]**, and **SCPP[10]** were investigated in dilute dichloromethane (DCM) solutions. As shown in Fig. 3a, the UV-vis absorption spectra of **SCPPs** exhibited broad absorption bands in the region of 280-480 nm...”

Biexponential fits to the emission lifetime data show that two components are present in ratios of ~80:20. It would be useful to show the expansion of the data, the biexponential fits, and the residuals. In addition, the 520 nm data are questionable due to the fast decay and similarity to the instrument response. A biexponential fit may not be sufficient. The authors may need to use a deconvolution of a Gaussian + a biexponential to accurately measure their data.

Response: We thank this reviewer very much for pointing out these. As suggested, we have showed the expansion of the data, the biexponential fits, and the residuals. And we have used a deconvolution of a Gaussian + a biexponential fit to remeasure some data as this reviewer suggested in the revised version. (see following Fig. 6a-c, Supplementary Figs. 26-27, and Supplementary Table 7).

Fig. 6 Emission lifetimes and TRES for SCPP[10,8]. a-c Emission lifetimes for SCPP[10,8]. **d** Time-resolved emission spectrum for SCPP[10,8].

Supplementary Figure 26. Emission lifetimes for SCPP[10,9].

Supplementary Figure 27. Emission lifetimes for SCPP[10].

Supplementary Table 7. Biexponential fitting results of PL decays for SCPPs.

Sample	wavelength	τ_1 (10^{-9} s)	τ_2 (10^{-9} s)	A_1 (%)	A_2 (%)	τ_{avg} (10^{-9} s)
SCPP[10]	390 nm	0.69	2.05	67.43	42.57	1.13
	470 nm	1.29	3.79	50.72	49.28	2.52
	520 nm	0.20	2.04	70.19	29.81	0.75
SCPP[10,9]	394 nm	0.94	2.75	81.18	18.82	1.28
	472 nm	1.15	3.14	44.89	55.11	2.25
	528 nm	0.16	2.78	63.59	36.41	1.11
SCPP[10,8]	392 nm	0.80	2.38	91.78	8.22	0.93
	467 nm	0.69	2.96	21.36	78.64	2.48
	545 nm	0.21	3.51	61.74	38.26	1.47

Overall, we appreciate the reviewers very much for these comments and suggestions to help us to improve our manuscript.

REVIEWER COMMENTS

Reviewer #1 (Remarks to the Author):

The authors have addressed most of my and the other reviewers comments. However, there are some important considerations left that I would like to see addressed before publication.

First, I suggest including the structure of SCPP[10] in figure 1 as well.

Interpreting the TCSPC on p16. rows 276-280. The authors write “ SCPPS showed a short-lived species and a long-lived species for all monitored emissions, suggesting that two relaxation pathways are involved in the decay processes.” This statement should be clarified, as a bi-exponential decay (as the authors are referring to) would not mean that two relaxation pathways are involved from the same excited state, but rather that two different excited states (or two distinct populations of the same excited states) with different radiative lifetimes are involved. Perhaps this is what the authors are meaning, but it can be misinterpreted. This also comes back to their discussion on row 388, where they state that two possible transitions are involved in the S2 and S1 emission, do they mean 2 transitions, one from each S2 and S1 or 4 transitions, 2 from each state? I would again argue that it indicates two possible populations, which is in line with their argument of 2 different conformers.

In relation to the conformers, it is interesting that in dilute PMMA films no anti-Kasha behavior is observed, which could imply that rotation/vibrational motion is required for the observed photophysical behavior in solution. Did the authors also observe bi-exponential emission in the PMMA films? Another change from dilute solution to PMMA films is the polarity of the media. Have the authors studied the emission of the SCPPs in various solvents, other than DCM, does the anti-Kasha emission change with polarity? This could help clarify the role of charge-transfer states in the photophysical behaviour.

Also, as response to reviewer 3 the authors claim to have used a gaussian deconvolution (why not IRF deconvolution directly as the IRF is available?) However, the plotted fits in figures 6, S26-s27 still seems to be from tail-fit method? Is that so? The authors could also consider plotting the TCSPC data with a log-y axis as it is often easier to distinguish the goodness of fit in that way.

Further, the authors do not mention what wavelength of excitation is used for the TCSPCS and TRES measurements in the main text.

Additionally, the times referred to in the TRES image 6d (e.g. 4.86-10.3 ns) I assume is related to the time window in figures 6a-c, and not the time after the pulse (as stated in the main text p16 row 289). It would make more sense to relate the time to the pulse, as it then also relates to what spectra are related to the short and long time components in the TCSPCS fitting.

Reviewer #3 (Remarks to the Author):

I thank and congratulate the authors for the significant efforts they have made to respond to my comments and to those of the other reviewers. These changes have improved the data presentation and interpretation, and I can recommend publication without further revision.

Point-by-point response to reviewer's comments:

Reviewer #1 (Remarks to the Author):

The authors have addressed most of my and the other reviewers comments. However, there are some important considerations left that I would like to see addressed before publication.

Response: We thank this reviewer very much for these positive comments and helpful suggestions to help us improve our work.

First, I suggest including the structure of **SCPP[10]** in figure 1 as well.

Response: Thank this reviewer very much for pointing out this. We have added the structure of **SCPP[10]** in the revised manuscript (see following Figure 1c).

Fig. 1 Schematic representation. **a** Structure of azulene. **b** Structures of thiophosgene (left) and xanthione (right). **c** Structures of heterogeneous bisnanohoops **SCPP[10,8]** (left), **SCPP[10,9]** (middle) and homogeneous bisnanohoop **SCPP[10]** (right). **d** Anti-Kasha emission of azulene. **e** S_2 , S_1 dual excited-state emissions of **SCPPs** bisnanohoops.

Interpreting the TCSPC on p16. rows 276-280. The authors write “ SCPPs showed a short-lived species and a long-lived species for all monitored emissions, suggesting that two relaxation pathways are involved in the decay processes.” This statement should be clarified, as a bi-exponential decay (as the authors are referring to) would not mean that two relaxation pathways are involved from the same excited state, but rather that two different excited states (or two distinct populations of the same excited states) with different radiative lifetimes are involved. Perhaps this is what the authors are meaning, but it can be misinterpreted. This also comes back to their discussion on row 388, where

they state that two possible transitions are involved in the S₂ and S₁ emission, do they mean 2 transitions, one from each S₂ and S₁ or 4 transitions, 2 from each state? I would again argue that it indicates two possible populations, which is in line with their argument of 2 different conformers.

Response: We thank this reviewer very much for pointing out this. We agree with this reviewer's viewpoint that “The bi-exponential decay of SCPPs indicated that two populations of the same excited states with different radiative lifetimes are involved in the decay processes”. We have corrected these misinterpretations, including the discussion on row 388, where we also mean that two populations are involved in the S₂ and S₁ emissions.

The related description on page 16, row 282, has been revised as follows: “SCPPs showed a short-lived and a long-lived species for all monitored emissions, suggesting that two populations are involved in the decay processes.”

The related description on page 22, rows 394-396, has been revised as follows: “The bi-exponential decays indicate that there are two possible populations with different radiative lifetimes involved in the S₂ and S₁ emissions.”

In relation to the conformers, it is interesting that in dilute PMMA films no anti-Kasha behavior is observed, which could imply that rotation/vibrational motion is required for the observed photophysical behavior in solution. Did the authors also observe bi-exponential emission in the PMMA films? Another change from dilute solution to PMMA films is the polarity of the media. Have the authors studied the emission of the SCPPs in various solvents, other than DCM, does the anti-Kasha emission change with polarity? This could help clarify the role of charge-transfer states in the photophysical behaviour.

Response: We thank this reviewer very much for this comment. According to this reviewer's suggestion, we have measured the emission lifetimes of SCPPs in the PMMA films using Steady-state/Lifetime Spectrofluorometer-Fluorolog-3-Tau upon excitation at 375 nm. These results also showed bi-exponential emissions in the PMMA films.

Fig. 1 a-c Emission lifetimes for SCPPs in the PMMA films (10 wt%).

In addition, we also studied the photophysical behavior of SCPPs in solvents with different polarities. Due to the poor solubility of SCPPs in common organic solvents,

we can only measure their emission spectra in a limited number of solvents with different polarities. The order of the polarities of these solvents is : Toluene < DCM < THF < MeCN < DMSO. These spectral results showed that **SCPPs** exhibit multiple emissions in all solvents, and photophysical behaviors of **SCPPs** are not directly related to solvent polarity.

Fig. II (a-c) Emission spectra of **SCPPs** in solvents of different polarities under excitations of 360 nm ($c = 5.0 \times 10^{-6}$ M). **(d-e)** Fluorescence photographs of **SCPPs** in solvents of different polarities (under 365 nm UV light).

Also, as response to reviewer 3 the authors claim to have used a gaussian deconvolution (why not IRF deconvolution directly as the IRF is available?) However, the plotted fits in figures 6, S26-s27 still seems to be from tail-fit method? Is that so? The authors could also consider plotting the TCSPC data with a log-y axis as it is often easier to distinguish the goodness of fit in that way.

Response: Thank this reviewer very much for pointing out this. The Gaussian deconvolution involves the processing of IRF, and some data are not used deconvolution because IRF is available. In the last manuscript, we performed Gaussian deconvolution on all the data, but only corrected the data with better fitting results (Emission lifetimes of **SCPP [10]** at 470 nm and 520 nm) in the Supplementary Figure 27 and Supplementary Table 7. However, we ignored the correction of fitting curves in Supplementary Figure 27 b-c. In the revised manuscript, we have corrected the fitting curves in Supplementary Figure 27 b-c using Gaussian deconvolution, which has been modified as follows (see following Supplementary Figure 27 b-c):

Supplementary Figure 27. Emission lifetimes for **SCPP[10]**.

Thank this reviewer for suggestions on plotting the TCSPC data with a log-y axis. Because the current TCSPC data are accurate and reliable, we decided not to make further modifications.

Further, the authors do not mention what wavelength of excitation is used for the TCSPCS and TRES measurements in the main text.

Response: Thank the reviewer for pointing out this. We mentioned what wavelength of excitations are used for the TCSPC on page 21, rows 380-383: “The time-resolved spectra were recorded using a PL lifetime spectrometer (Edinburgh Instruments, LifeSpec-II) based on a time-correlated single photon counting (TCSPC) technique with excitations of 375 nm and 315 nm picosecond laser.” According to the reviewer's suggestion, we have added detailed information on excitations used for the TCSPCS and TRES measurements in the main text.

The related description (*page 16, in the first paragraph*) has been revised as follows: “Emission lifetimes of **SCPPs** at ~390 nm were measured upon excitation at 315 nm, and emission lifetimes at ~470 nm, 520 nm, 528 nm and 545 nm were measured upon excitation at 375 nm.

The related description (*page 17, in the first paragraph*) has been revised as follows: The TRES of **SCPP[10,8]** was measured upon excitation at 375 nm.”

Additionally, the times referred to in the TRES image 6d (e.g. 4.86-10.3 ns) I assume is related to the time window in figures 6a-c, and not the time after the pulse (as stated in the main text p16 row 289). It would make more sense to relate the time to the pulse, as it then also relates to what spectra are related to the short and long time components in the TCSPCS fitting.

Response: We thank this reviewer for this comment and we agree with it. We have deleted the description of “after the laser pulse” and corrected the corresponding description in the revised manuscript.

The related description (*page 17, in the first paragraph*) has been revised as follows: “At short delay time (4 .856 ns), the spectrum showed a narrow emission peak at 390 nm and a broad emission band maximized at 550 nm with a small shoulder...”

Overall, we appreciate the reviewers very much for these comments and suggestions to help us to improve our manuscript.

Reviewer #3 (Remarks to the Author):

I thank and congratulate the authors for the significant efforts they have made to respond to my comments and to those of the other reviewers. These changes have improved the data presentation and interpretation, and I can recommend publication without further revision.

Response: We thank this reviewer very much for the positive comments and great support to help us improve our work.

REVIEWERS' COMMENTS

Reviewer #1 (Remarks to the Author):

The authors have addressed all of my scientific concerns. However there seems to be a slight confusion about my final comment.

The comment read as: "Additionally, the times referred to in the TRES image 6d (e.g. 4.86-10.3 ns) I assume

is related to the time window in figures 6a-c, and not the time after the pulse (as stated in the main text p16 row 289). It would make more sense to relate the time to the pulse, as it then also relates to what spectra are related to the short and long time components in the TCSPCS fitting."

Their Response: We thank this reviewer for this comment and we agree with it. We have deleted the description of "after the laser pulse" and corrected the corresponding description in the revised manuscript.

The related description (page 17, in the first paragraph) has been revised as follows:

"At short delay time (4.856 ns), the spectrum showed a narrow emission peak at 390 nm and a broad emission band maximized at 550 nm with a small shoulder..."

My concern is that the changed text does not make a difference. At short delay times usually also refers to delay after excitation. The delay in the window is arbitrary and the 4.856 ns is setup specific and says nothing of what time the spectra is recorded in relation to the excitation. For example with lifetimes $\ll 4$ ns one would interpret 4.856 ns as one of the final spectra recorded during the decay. I suggest the authors rescale the time axis to center the excitation to 0 ns and refer to time delay from this point. Or at least specify at what time the pulse arrives.

This is a minor issue and further review is not necessary, but it should be corrected before publication.

Point-by-point response to reviewer's comments:

Reviewer #1 (Remarks to the Author):

The authors have addressed all of my scientific concerns. However there seems to be a slight confusion about my final comment.

The comment read as: "Additionally, the times referred to in the TRES image 6d (e.g. 4.86-10.3 ns) I assume is related to the time window in figures 6a-c, and not the time after the pulse (as stated in the main text p16 ro5w 289). It would make more sense to relate the time to the pulse, as it then also relates to what spectra are related to the short and long time components in the TCSPCS fitting."

Their Response: We thank this reviewer for this comment and we agree with it. We have deleted the description of "after the laser pulse" and corrected the corresponding description in the revised manuscript.

The related description (page 17, in the first paragraph) has been revised as follows: "At short delay time (4.856 ns), the spectrum showed a narrow emission peak at 390 nm and a broad emission band maximized at 550 nm with a small shoulder..."

My concern is that the changed text does not make a difference. At short delay times usually also refers to delay after excitation. The delay in the window is arbitrary and the 4.856 ns is setup specific and says nothing of what time the spectra is recorded in relation to the excitation. For example with lifetimes $\ll 4$ ns one would interpret 4.856 ns as one of the final spectra recorded during the decay. I suggest the authors rescale the time axis to center the excitation to 0 ns and refer to time delay from this point. Or at least specify at what time the pulse arrives.

This is a minor issue and further review is not necessary, but it should be corrected before publication.

Response: We thank this reviewer for this comment. According to this reviewer's suggestion, we have rescaled the time axis to center the excitation to 0 ns and refer to time delay from this point (see following Fig. 6d).

Fig. 6 Emission lifetimes and TRES for SCPP[10,8]. a-c Emission lifetimes for SCPP[10,8]. **d** Time-resolved emission spectrum for SCPP[10,8].

Overall, we appreciate this reviewer very much for these comments and suggestions including all previous comments to help us to improve our manuscript.